# Characteristics of Dutch ED patients and their journey through the acute care chain: A province-wide flash-mob study

Lieke Claassen[1]*, Patricia Maria Stassen[2], Thimo Jozef Theresia Boumans[1], Dennis Gerard Barten[3], Marjolein Nel Tinie Kremers[4¤], Anne Maria Elisa Hermans[3], Noortje Zelis[2], Jochen Wilco Lennert Cals[5], Gideon Hubertus Petrus Latten[1]

1 Department of Emergency Medicine, Zuyderland Medical Centre, Heerlen, The Netherlands, 2 Section Acute Medicine, Division General Medicine, Department of Internal Medicine, CARIM School of Cardiovascular Diseases, Maastricht University, Maastricht, The Netherlands, 3 Department of Emergency Medicine, VieCuri Medical Centre, Venlo, The Netherlands, 4 Department of Emergency Medicine, Sint Jans Gasthuis, Weert, The Netherlands, 5 Department of Family Medicine, Care and Public Health Research Institute (CAPHRI), Maastricht University, Maastricht, The Netherlands

¤ Current address: Department of Emergency Medicine, Erasmus University Medical Centre, Rotterdam, The Netherlands
* l.claassen@zuyderland.nl

## Abstract

### Background

Insight in characteristics and journey of patients in the Acute Care Chain (ACC) who visit the Emergency Department (ED) is lacking. Existing studies focus on prespecified (time-sensitive) complex conditions like major trauma and stroke, and on the hospital phase. This study provides a representative overview of adult ED patients and their journey through the ACC with focus on differences between those with and without prespecified complex conditions.

### Methods

A prospective 72-hour flash-mob study was conducted in 2022 across all six EDs in the province of Limburg, the Netherlands, encompassing unselected adult patients. Baseline characteristics, journey, and time within ACC were collected. Patients with a prespecified complex condition (stroke, myocardial infarction, aortic syndrome and major trauma) were compared to those without.

### Results

Out of 794 adult ED patients, 585 (73.7%) were included. Patients reported symptoms for a median of 1 day (IQR 0-4) before their ED visit; 56.3% encountered ≥1 healthcare provider. General practitioners referred 56.1% of patients, and emergency medical services transported 32.9%. The median time in ACC was 5.0 hours (IQR 3.5-6.9), with 3.0 hours (IQR 2.0-4.2) spent in the ED. The three most prevalent presenting complaints were minor trauma (28.5%), abdominal pain (14.6%) and dyspnoea (12.5%), while 9.3% presented

**Data availability statement:** Data is available via the corresponding author Lieke Claassen l.claassen@zuyderland.nl or the Scientific Research Education office (Bureau Wetenschappelijk Onderwijs) BWO@zuyderland.nl upon reasonable request. All patients provided informed consent for the collection and use of their data for the purposes of this study. However, consent for public data sharing was not obtained. This approach was reviewed and approved by the local medical ethics committee in accordance with applicable ethical guidelines and regulations.

**Funding:** The SGO fund, a Dutch research fund for Emergency Physicians, provided a "Kick Start fund" of 1000 euro's. The funders had no role in study design, data collection and analysis, decision to publish, or preparation of the manuscript.

**Competing interests:** The authors have declared that no competing interests exist.

**Abbreviatons:** ACC, Acute Care Chain;CRF, Case Report Form;CT, Computed Tomography;ECG, Electrocardiogram;ED, Emergency Department;ED-LOS, Emergency Department Length Of Stay;EMS, Emergency Medical Services;GP, General Practitioner;GPC, General Practitioner Cooperative;ICU, Intensive Care Unit;IQR, Interquartile Range;LP, Lumbar Puncture;Med, Median;METC-Z, Medical Ethics Committee of Zuyderland;MRI, Magnetic Resonance Imaging;MTS, Manchester Triage System;NTS, Netherlands Triage Standard;UK, United Kingdom;SD, Standard Deviations;STROBE, STrengthening the Reporting of OBservational in Epidemiology.

with prespecified complex conditions. Patients with a prespecified complex condition were more often triaged highly urgent (53.6% vs 13.9%, p < 0.001) and received a complex work-up (79.6% vs 41.2%, p < 0.001).

## Conclusion

In our province-wide study, ED patients had symptoms for 1 day and over half of them already contacted a healthcare provider. Time in ACC was 5 hours, with a substantial proportion of time spent prehospitally. Prespecified complex conditions accounted for less than 1 in 10 ED patients. These findings highlight that, to optimise care and care policy, it is essential to examine the entire ACC for unselected patients.

## Trial registration

ClinicalTrials.gov NCT06079099

## Introduction

Prehospital and hospital healthcare providers within the acute care chain (ACC) work closely together to provide appropriate care at the appropriate time. In this ACC, the Emergency Department (ED) serves as a pivotal "station" to which patients with a variety of symptoms travel through various routes, e.g., after referral by a General Practitioner (GP) and/or transportation by Emergency Medical Services (EMS). Traditionally, most research within the ACC focuses on patients presenting at specific locations such as the ED, GP or EMS, and on specific medical conditions, with limited exploration of the prehospital phase [1–3]. Since the quality of care provided by the consecutively involved professionals determines patient outcomes, the entire ACC should be subject of research.

In patients with prespecified complex conditions, such as stroke, myocardial infarction, aortic syndrome and major trauma, the development and implementation of effective care pathways have led to better patient outcomes [2–5]. Although it is likely that more acute conditions could benefit from such care pathways, data on these patients is lacking [6]. Possible explanations for this knowledge gap include the fact that approximately 65% of ED patients are referred with a complaint or symptom instead of a presumptive diagnosis and the atypical presentation of some conditions (e.g., sepsis) [7,8]. Truly improving care for patients can therefore be challenging, and a baseline overview of the ACC and his patients is crucial.

In this prospective flash-mob study, we therefore aimed to gain insight into the characteristics and the journey through the ACC of all adult ED patients in an entire Dutch province. We specifically focused on the composition of the population in the ACC, their route and the time spent there. In addition, we focused on the differences between patients with the presumptive diagnosis of a prespecified complex condition and other ED patients.

## Methods

### Study design and setting

In this prospective flash-mob study, all adult patients who presented at one of the six EDs in the province of Limburg, the Netherlands, were included within a 72-hour consecutive time period (Thursday April 21st 2022 at 8 AM – Sunday April 24th 2022 at 8 AM). The flash-mob method is a new way of conducting prospective research allowing relatively simple – but clinically relevant – questions to be answered in a short period of time [9].

Patients were included in all six EDs that provide care to over 1 million citizens: VieCuri Medical Centre Venlo, Sint Jans Gasthuis Weert, Laurentius Hospital Roermond, Maastricht University Medical Centre, Zuyderland Medical Centre Heerlen and Zuyderland Medical Centre Sittard-Geleen. These comprise teaching, non-teaching and university hospitals, with a combined annual census of approximately 135,000 patients. In 5 hospitals, a separate cardiac emergency unit is present, from which no patients were included in this study. In these units, stable cardiac patients are treated. Unstable cardiac patients are mainly presented to the ED.

In the Netherlands, GPs serve as a gatekeeper, providing the first step in emergency care 24/7 in their practices during office hours and in general practitioner cooperatives (GPCs) during out-of-hours. EDs and EMS provide care for the minority of patients; those who requires specialised care [10–12]. Patients are usually referred by a GP, a medical specialist, or directly transferred by an ambulance in order to access the ED. Compared to other healthcare systems, self-referrals are relatively uncommon [12,13].

## Patients

Eligible for inclusion were all adults (≥ 18 years) presenting to the ED. Written informed consent had to be provided by the patient or their legal representative prior to inclusion. Exclusion criteria were a language barrier and second presentation within the study period, since revisits were one of the outcome measures. A research team consisting of students and/or physicians was present 24/7 at each ED to include patients. When immediate inclusion in the study was not possible (e.g., due to crowding or being too severely ill), the patient could be asked for participation by the research team in a later stage (<24 hours).

## Data collection

Patient data were collected by well-trained researchers under the supervision of local ambassadors through a questionnaire completed by the professional, utilising information from medical records (including hospital files, referral letters and EMS notes) as well as a questionnaire filled out by the patient. Together, these sources comprised a Case Report Form (S1 Questionnaire). Data were collected on demographic variables, the patient journey and the ACC.

We used the strengthening the Reporting of Observational Studies in Epidemiology guidelines to report this study (STROBE) [14]. This study was reviewed and approved by the medical ethics committee of Zuyderland (METC-Z nr. 20210142) and boards of directors of all participating hospitals.

## Definitions and variables of interest

**Demographics.** Age, highest level of education and current living situation were retrieved.

**Acute care chain.** The ACC was divided into four phases: the pre-referral, referral, ED and post-ED phase. In each phase several variables were collected (for details and definitions see Table 1).

**Patient journey.** Six possible patient journeys were identified within the ACC:

- GP referral + own transportation (GP + own);

- GP referral + EMS transportation (GP + EMS);

- Calling 112 (national emergency number) + EMS transportation (112 + EMS);

- Other healthcare provider referral + own transportation (other + own);

- Other healthcare provider referral + EMS transportation (other + EMS);

- Self-referral.

**Table 1. Definitions of variables collected in the Acute Care Chain.**

| | |
|---|---|
| **Pre-referral phase** *(Primary onset of symptoms until referring contact)* | |
| **Symptom duration (days)** | Duration of symptoms prior to index ED visit, during current disease episode |
| **Prior contact with a healthcare provider** | Prior contact with healthcare provider within this disease episode regarding the same complaint, before the referral contact |
| **Prescribed medication** | Medication prescribed this disease episode prior to current ED visit |
| **Referral phase** *(Referring contact until arrival at the ED)* | |
| **Referral contact** | Contact during which the patient was referred to the ED |
| **Part of the day of referring contact with healthcare provider** | Day: 8.00 am – 17.00 pm<br>Evening: 17.00 pm – 23.00 pm<br>Night: 23.00 pm – 8.00 am |
| **Referred by** | GP, EMS, other health care provider or self-referral |
| **Urgency level GPC** | According to NTS: U1 (life threatening) - U5 (can wait until next day) [29]. U1, U2, U3 were combined as " highly urgent" and U4 and U5 as "non-urgent" |
| **Mode of transportation** | By EMS or own transportation |
| **Urgency level EMS** | A1 (emergency, within 15 minutes), A2 (urgent, within 30 minutes), B (non-urgent) |
| **ED phase** *(Arrival at the ED until ED discharge)* | |
| **Index ED visit** | First ED visit within study period |
| **Referred specialty** | The specialty patients are referred to by the referring healthcare provider |
| **Treating specialty** | The specialty that eventually treated and discharged or admitted the patient |
| **Urgency level ED** | ED urgency levels were determined according to the MTS [30]. We combined red and orange triage urgencies as 'highly urgent', and yellow, green and blue as 'urgent' |
| **Presenting symptoms** | Symptom for which the patient primarily presented |
| **Prespecified complex condition** | Presumptive diagnosis of stroke, myocardial infarction, aortic syndrome or major trauma |
| **ED work-up** | Ancillary investigations, defined as:• Simple: laboratory testing, ECG, conventional radiography<br>• Complex: CT, ultrasound, MRI, lumbar puncture, abdominal paracentesis, endoscopy and<br>• consultation of other specialties |
| **Length of stay in the ED (ED-LOS)** | Time between ED arrival and ED departure |
| **Post-ED phase** *(30-day follow-up period after ED discharge)* | |
| **Post-ED disposition** | Discharge from hospital, admission to hospital, transfer to other hospital |
| **Adverse outcomes** | 30-day all-cause mortality<br>30-day revisit |

**Time in ACC.** The time in ACC was defined as the time between the moment of contacting the referring healthcare provider (physical or telephonic consultation) and the time the patient left the ED.

**Prespecified complex conditions.** In the Netherlands, specific care pathways have been established for patients with a suspected stroke, myocardial infarction, aortic syndrome (including aortic dissection and ruptured aortic aneurysm) or major trauma. We defined these conditions, which were retrieved from referral notes, as prespecified complex conditions.

## Statistical analysis

We performed descriptive analyses of patient characteristics and of the variables of the four phases, both overall and for the different patient journeys.

Patients with prespecified complex conditions were compared with those with non-prespecified complex conditions regarding demographics, several variables of the different phases, adverse outcomes and time in ACC. For the group of non-included eligible patients, limited baseline data (i.e., sex, age and ED urgency level) was collected to identify possible differences between included and non-included patients.

Continuous variables were reported as means with standard deviations (SD) when normally distributed and compared using Students' T test, or medians with interquartile ranges (IQRs) compared using Mann Whitney U test, Kruskal Wallis test or One way Anova, when not normally distributed. Testing for a normal distribution was performed both visually (histogram, boxplot) and statistically (skewness, kurtosis). Categorical variables were reported as absolute numbers and as in case of missing data, valid percentages were calculated. Comparisons were made using chi-square or Fisher exact tests. Differences were considered significant when $p < 0.05$. A sample size calculation was not performed due to the explorative and descriptive nature of the study. All statistical analyses were performed using IBM SPSS statistical software version 28 (Chicago, Illinois, USA).

## Patient and public involvement

Patients or the public were not involved in the design, or conduct, or reporting, or dissemination of this research.

## Results

### Participants

During the inclusion period, 794 adult patients visited the six EDs (Fig 1). Of these, 728 (91.7%) were asked for participation and in total, 585 (73.7%) patients were included after obtaining informed consent. Due to missing data, we excluded 2 additional patients, resulting in 583 patients (73.4%) for final analysis.

### Baseline characteristics

Baseline characteristics are shown in Table 2. The median age was 65 years (IQR 47-76) and 49.7% of patients were female. Most (81.5%) patients were living independently.

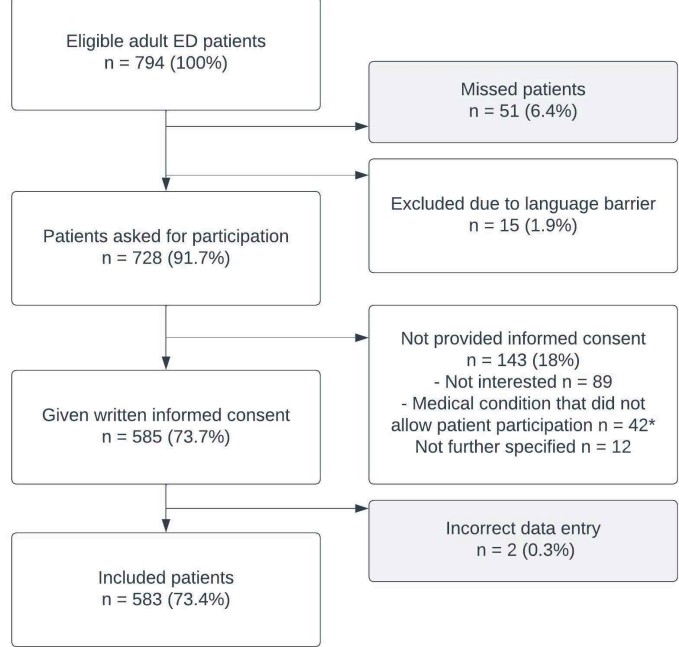

**Fig 1. Flowchart of study population.** *Acute condition or cognitive impairment.

**Table 2. Baseline characteristics.**

| Demographics (n = 583) | n (%) |
|---|---|
| Sex | |
| Male | 293 (50.3%) |
| Female | 290 (49.7%) |
| Age in years, median (IQR) | 65 (47-76) |
| 18-49 years | 153 (26.2%) |
| 50-64 years | 135 (23.2%) |
| 65-79 years | 192 (32.9%) |
| ≥80 years | 103 (17.7%) |
| Highest level of education attained by participant [a] | |
| No education | 11 (1.9%) |
| Primary school | 53 (9.1%) |
| Secondary School | 200 (34.3%) |
| Vocational training | 164 (28.2%) |
| University | 154 (26.5%) |
| Living situation | |
| Independent living | 475 (81.5%) |
| Community dwelling with domiciliary care | 76 (13.0%) |
| Assisted living | 6 (1.0%) |
| Nursing home | 20 (3.5%) |
| Other | 6 (1.0%) |
| Roommates [b] | |
| Yes | 440 (76.0%) |
| No | 139 (24.0%) |

Values are n (%) for ordinal variables and median (IQR) for continues variables. Abbreviations: IQR – interquartile range.

[a]One missing.

[b]Four missing.

## ED patients in the acute care chain

The median duration of symptoms before ED visit was 1 day (IQR 0-4) (Table 3). In the phase preceding the ED visit, 262 (44.9%) patients consulted a healthcare provider at least once and 193 (33.1%) had already been prescribed medication.

On the day of the ED visit, GPs had referred 327 (56.1%) of patients and 192 (32.9%) were transported by EMS. Most patients (n = 387, 66.4%) were referred during daytime.

At the ED, 102 (17.8%) patients were triaged as highly urgent. Three of the presenting complaints dominated and collectively accounted for 55.6% of all cases: minor trauma (n = 166, 28.5%), abdominal pain (n = 85, 14.6%) and dyspnoea (n = 73, 12.5%). In total, 54 (9.3%) patients presented with the presumptive diagnosis of a prespecified complex condition. Nearly all patients (91.7%) underwent ancillary investigations, 48.6% of these receiving a complex work-up.

After the ED visit, 261 (44.8%) patients were admitted to the hospital. In total, within 30 days, 27 patients (4.6%) died and 77 (13.2%) revisited the ED, and follow-up was complete for all patients.

## Patient journeys and time in the acute care chain

The most common journey (47.2%) consisted of referral by a GP and own transportation to the ED (Fig 2). Self-referrals accounted for 6.3% and represented the least common journey.

Table 3. Patient characteristics of the different phase of the Acute Care Chain (n = 583).

| Pre-referral phase | n (%) or median (IQR) |
|---|---|
| Symptom duration (days) | 1 (0-4) |
| Prior contact with a healthcare provider | 262 (44.9%) |
| Prescribed medication | 193 (33.1%) |
| **Referral phase** | |
| Part of the day of referral (n = 559)[a] | |
| Day (8-17h) | 387 (69.2%) |
| Evening (17-23h) | 120 (21.5%) |
| Night (23-8h) | 52 (9.3%) |
| Referred by | |
| GP | 327 (56.1%) |
| EMS | 129 (22.1%) |
| Other healthcare provider | 90 (15.5%) |
| Self-referral | 37 (6.3%) |
| Urgency level GPC (n = 83)[b] | |
| Highly urgent | 70 (84.3%) |
| Non-urgent | 13 (15.7%) |
| EMS transportation | 192 (32.9%) |
| Urgency level EMS (n = 184)[c] | |
| A1 (highest) | 39 (21.2%) |
| A2 | 131 (71.2%) |
| B | 14 (7.6%) |
| **ED phase** | |
| Referred specialty (n = 546) | |
| Surgery | 117 (21.4%) |
| Internal medicine | 109 (20.0%) |
| Emergency medicine | 83 (15.2%) |
| Orthopaedics | 56 (10.2%) |
| Neurology | 67 (12.3%) |
| Pulmonology | 60 (11.0%) |
| Gastroenterology | 17 (3.1%) |
| Cardiology | 17 (3.1%) |
| Urology | 13 (2.4%) |
| Other | 7 (1.3%) |
| Urgency level ED (n = 574)[d] | |
| Highly urgent | 102 (17.8%) |
| Urgent | 472 (82.2%) |
| Presenting complaints | |
| Minor trauma | 166 (28.5%) |
| Abdominal pain | 85 (14.6%) |
| Dyspnoea | 73 (12.5%) |
| General malaise | 42 (7.2%) |
| Fever | 27 (4.6%) |
| Extremity complaints | 26 (4.5%) |
| Neurological complaints | 25 (4.3%) |
| Syncope/ palpitations | 17 (2.9%) |
| Skin symptoms | 15 (2.6%) |
| Urological complaints | 10 (1.7%) |

*(Continued)*

**Table 3.** (Continued)

| Pre-referral phase | n (%) or median (IQR) |
|---|---|
| Complaints after treatment | 10 (1.7%) |
| Allergic reaction | 6 (1.0%) |
| Hypertension | 6 (1.0%) |
| Other | 21 (3.6%) |
| Prespecified complex conditions | 54 (9.3%) |
| ED work-up | |
| *Ancillary investigations* | 537 (91.7%) |
| Simple work up | 276 (51.4%) |
| Complex work up | 261 (48.6%) |
| **Post-ED phase** | |
| Discharge destination | |
| Home | 322 (55.2%) |
| Hospital admission | 261 (44.8%) |
| General Ward | 219 (83.9%) |
| Medium care or ICU | 36 (13.8%) |
| Transfer to other hospital | 6 (2.3%) |
| Adverse outcome: | |
| 30-day mortality | 27 (4.6%) |
| Revisit in 30-days (n = 562) | 77 (13.2%) |

Values are n (%) for ordinal variables and median (IQR) for continues variables, median (IQR).

Abbreviations: ED – emergency department; GP – general practitioner; EMS – emergency medical services; ICU – Intensive Care Unit.

[a]24 missing.

[b]For 83 patients urgency levels at the GPC were registered.

[c]Eight missing.

[d]Nine missing.

For 557 (95.5%) patients, the time in ACC could be calculated. The median time in ACC was 300 minutes (IQR 212-417) (5 hours), of which median 87 minutes (IQR 46-176) in the referral phase, and median 183 minutes (IQR 121-253) in the ED phase (ED-LOS) (Fig 2).

The longest patient journey was after referral by other healthcare providers (not by GP) and consecutive transportation by EMS (median 376 minutes (IQR 305-609)). Patients who presented to the ED after self-referral experienced the shortest time in ACC (median 217 minutes (IQR 138-335)).

## Prespecified complex conditions

Of all patients, 54 (9.3%) were referred with a presumptive diagnosis of a prespecified complex condition (Table 4). These patients were older (68 vs. 65 years, p = 0.023) and less likely had prior contact with a healthcare provider before the referral contact (29.3% vs. 46.7%, p = 0.012), when compared to patients without a suspected prespecified complex condition. In addition, they were more often transported by EMS after calling 112 (56.9% vs. 18.3%, p < 0.001), and often with the highest EMS urgency (46.2% vs. 13.7%, p < 0.001). In the ED, they were more frequently triaged as highly urgent (53.6% vs. 13.9%, p < 0.001) and more often received a complex work-up (79.6% vs. 41.2%, p < 0.001).

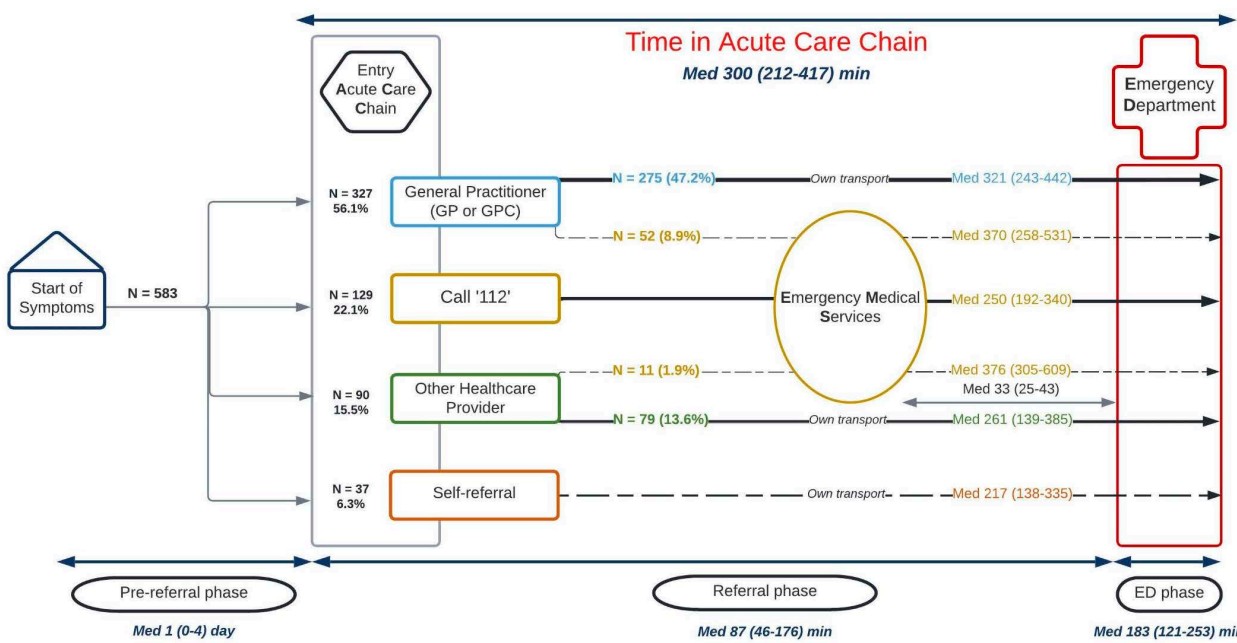

**Fig 2. Patient journeys in the Acute Care Chain and the time in Acute Care Chain.** Abbreviations: Med – median; ED – Emergency Department; GP – General Practitioner; GPC – General Practitioner Cooperative.

Regarding adverse outcomes, 30-day mortality was higher in patients with prespecified complex conditions, although this was not statistically significant (n = 5, 9.3% vs. n = 22, 4.2%, p = 0.089). No significant differences were found in 30-day revisits nor in the combined adverse outcome (30-day mortality or revisit). Patients with a prespecified complex condition also spent significantly less time in ACC when compared to patients without one (median 231 vs. 305 minutes, p < 0.001).

## Comparison of included with non-included patients

When comparing the 583 included with 211 non-included patients, we found no significant differences in sex and age, but included patients were less often triaged as highly urgent (17.5% vs. 24.7%, p = 0.013) (S2 Table).

## Discussion

In this prospective flash-mob study, we investigated the patient characteristics, journey and time in ACC of all adult ED patients in the province of Limburg, the Netherlands, over a 72-hour period. We found that patients had symptoms for a median of 1 day, the majority of patients were referred by a GP, and in half of the cases patients contacted a healthcare provider prior to their referral contact or before their self-presentation at the ED. This study showed that ED patients are a highly heterogeneous population with regard to their journey and their presenting complaints, with only 1 in 10 presenting with the presumptive diagnosis of a prespecified complex condition. In total, adult ED patients spend 5 hours in the ACC of which about one-third in the prehospital phase.

To the best of our knowledge, this is the first study investigating the characteristics, journey and time in ACC of adult ED patients. The heterogeneity of our ED population is illustrated by the finding that the three most frequently reported presenting complaints (minor trauma,

**Table 4. Comparison of patients with and without prespecified complex conditions (n = 583).**

| | Prespecified complex conditions (n = 54, 9.3%) | Non-prespecified complex conditions (n = 529, 90.7%) | |
|---|---|---|---|
| | n (%) or median (IQR) | n (%) or median (IQR) | p-value |
| Sex | | | 0.430 |
| Male | 32 (55.2%) | 261 (49.7%) | |
| Female | 26 (44.8%) | 264 (50.3%) | |
| Age | 68 (58-78) | 65 (46-76) | |
| **Pre-referral phase** | | | |
| Symptom duration (days) | 0 (0-1) | 1 (0-5) | <0.001 |
| Prior contact with a healthcare provider | 17 (29.3%) | 245 (46.7%) | 0.012 |
| Prescribed medication | 20 (34.5%) | 165 (32.9%) | 0.736 |
| **Referral phase** | | | |
| Patient journey | | | <0.001 |
| GP referral + own transport | 14 (25.9%) | 261 (49.3%) | |
| GP referral + EMS transport | 7 (13.0%) | 45 (8.6%) | |
| Calling 112 + EMS transport | 32 (59.3%) | 97 (18.3%) | |
| Other referral + own transport | 0 | 79 (14.9%) | |
| Other referral + EMS transport | 0 | 11 (2.1%) | |
| Self-referral | 1 (1.9%) | 36 (6.8%) | |
| GP referral | 23 (39.7%) | 304 (57.9%) | |
| EMS transport | 39 (72.2%) | 153 (28.9%) | <0.001 |
| Urgency level EMS | | | <0.001 |
| A1 | 18 (46.2%) | 21 (13.7%) | |
| A2 | 21 (53.8%) | 109 (71.9%) | |
| B | 0 | 14 (9.2%) | |
| Unknown | 0 | 8 (5.2%) | |
| **ED phase** | | | |
| Triage: highly urgent (red or orange) | 30 (53.6%) | 72 (13.9%) | <0.001 |
| ED work-up | | | <0.001 |
| Simple | 11 (20.4%) | 311 (58.8%) | |
| Complex | 43 (79.6%) | 218 (41.2%) | |
| **Post-ED phase** | | | |
| Adverse outcome (deceased or revisit) | 11 (20.4%) | 92 (17.4%) | 0.585 |
| 30-day mortality | 5 (9.3%) | 22 (4.2%) | 0.089 |
| 30-day revisit | 6 (11.1%) | 71 (13.4%) | 0.633 |
| **Time spent in ACC** | | | |
| Median time in ACC | 231 (161-319) | 305 (217-427) | <0.001 |

abdominal pain and dyspnoea) collectively accounted for half of the complaints, while the remaining half comprised at least ten other complaints. Other studies also illustrate a wide range of presenting complaints [15–17].

Furthermore, almost half of patients had prior contact with a healthcare provider at least once before the referral contact, which is in line with research in ED patients with sepsis [18]. Before referral, our patients had a median duration of symptoms of 1 day. Since no other study has investigated the entire population before, we can only compare these findings with data on specific subgroups. In a Canadian study, the median duration of symptoms was 0.8

days for patients with cardiac conditions and 4.0 days for pulmonary infections [19]. A study in stroke patients showed a mean of 15 hours, with 43.6% presenting within 4 hours [20]. The variance in symptom duration in these studies highlight the heterogeneity of the ED population and the variance in the prehospital phase.

Regarding the patient journey, our study showed a high proportion of patients being referred by healthcare providers. This is in line with another Dutch study regarding the proportion of patients referred by GP, by EMS and by other healthcare providers [10,12,13]. These healthcare providers appear to make accurate referral decisions, as further ancillary investigations were performed in over 90% of patients, and nearly half were admitted. This is a much higher admission rate than in the UK with about 29% of ED patients being admitted [21]. The involvement of different healthcare providers and the variation in referring professionals highlight that there is more than the ED phase and that the prehospital phase is worth investigating.

The median time in ACC was 5 hours, with about one-third spent in the prehospital phase. The longest routes were through a referral (by either GP or other professional) and subsequent transport by EMS. One could conclude that the involvement of prehospital healthcare providers contributes to this longer journey, but they also contribute to selection of ED patients. As there are no other studies reporting the total time in ACC, we can compare only our ED-LOS with other studies. Our ED-LOS of approximately three hours was similar or slightly longer compared to other Dutch studies (approximately 13 to 50 minutes) [13,22]. This may be due to our population being older than in the rest of the country [23]. More importantly, the entire time in ACC in our study was even shorter than the ED-LOS in other countries [24–26]. This short ED-LOS in Dutch studies is most likely a reflection of the strong primary care system, resulting in fewer self-referrals and fewer patients overall due to triage and treatment provided by the GP [12,13].

In our study, we specifically focused on patients presenting with the presumptive diagnosis of a prespecified complex condition. Only 1 in 10 patients presented with such a condition. This is an important finding since due to crowding and governmental decisions, healthcare is becoming more centralised [27]. Even these patients with a prespecified condition did not form a homogeneous cohort as we found only half of them arriving with the highest EMS and ED triage urgency. Likewise, 1 in 7 patients in the non-prespecified complex condition group were triaged highly urgent both by the EMS and the ED. Furthermore, 80% of the prespecified complex group received a complex work-up, but this was also the case in 40% of the non-prespecified complex group. These results highlight the importance of considering underlying critical conditions in all patients, especially given the high mortality observed in both groups. Since patients with prespecified complex conditions account for only 10% of ED patients, we recommend further investigations into unselected patients to enhance understanding of their diagnostic needs and care pathways.

Presenting the right patient in the right place at the right time and making sure that specialised acute care is always available for those in need, is a challenge in healthcare. We acknowledge the unique nature of the Dutch healthcare system with an important role for GPs. As a result of their gatekeeping role, we observed low rates of self-referrals in our study (6.3%), as well as in another Dutch study (14.9%) [10]. Patients in the ED usually are in need of specialised acute care considering the high admission rates and the extent of a complex work-up. GPs are able to handle about 80% of the acute care problems [28]. Although it is challenging to generalise our results due to differences with other care systems, we are optimistic that other nations could benefit from the experiences of the Dutch acute care organisation.

## Limitations

Despite being the first to investigate the patient journey and time in ACC of adult ED patients, and the strengths associated with its prospective design, our study has limitations. It

is possible that due to our flash-mob design, our population is not representative for the ED population (e.g., due to seasonal variations), however we selected both weekdays and weekend days.

Furthermore, we were unable to approach all eligible patients for inclusion despite the 24/7 presence of the research team and the opportunity to include patients within 24 hours after ED presentation. This was due to crowding in the ED, some patients leaving before inclusion could take place, or some patients requiring only nursing care (e.g., placement of bladder catheter). Sub-analysis comparing included versus non-included patients, revealed a significantly higher proportion of highly urgent patients in the non-included group, which may indicate that some patients with prespecified complex conditions were missed. Despite these limitations, we believe that this study provides a valuable first insight in ED patients in the ACC.

## Conclusion

In conclusion, this study showed who our ED patients are and delineates their journey through the ACC. Our study confirms the highly heterogeneous nature of ED patients, with only a small proportion presenting with a presumptive diagnosis of a prespecified complex condition. The majority of patients are referred by a GP, and a third is transported by EMS. Out of the total 5-hour duration in the ACC, about one-third is spent in the prehospital phase. This study highlights the importance of the phase prior to actual referral and ED visit. Further research needs to extend its focus beyond the ED and beyond specific conditions to optimise care and care policy.

## Supporting information

**S1 Questionnaire. Questionnaire patient and professional.**
(DOCX)

**S2 Table. Comparison between included and non-included eligible patients.** Values are n(%) for ordinal variables and median (IQR) for continues variables, median (IQR). Abbreviations: ED – emergency department. *2 missing in eligible group **p<0.05.
(DOCX)

## Acknowledgments

The authors thank Audrey Merry (Zuyderland MC) for her assistance with this article. We would also like to thank all the students and physicians that helped with the inclusion of the patients.

## Author contributions

**Conceptualization:** Lieke Claassen, Patricia Maria Stassen, Jochen Wilco Lennert Cals, Gideon Hubertus Petrus Latten.

**Data curation:** Lieke Claassen, Thimo Jozef Theresia Boumans, Dennis Gerard Barten, Marjolein Nel Tinie Kremers, Gideon Hubertus Petrus Latten.

**Formal analysis:** Lieke Claassen, Thimo Jozef Theresia Boumans.

**Funding acquisition:** Lieke Claassen.

**Investigation:** Lieke Claassen, Patricia Maria Stassen, Dennis Gerard Barten, Marjolein Nel Tinie Kremers, Anne Maria Elisa Hermans, Noortje Zelis, Gideon Hubertus Petrus Latten.

**Methodology:** Lieke Claassen, Patricia Maria Stassen, Gideon Hubertus Petrus Latten.

**Project administration:** Lieke Claassen, Thimo Jozef Theresia Boumans.

**Supervision:** Patricia Maria Stassen, Noortje Zelis, Jochen Wilco Lennert Cals, Gideon Hubertus Petrus Latten.

**Visualization:** Gideon Hubertus Petrus Latten.

**Writing – original draft:** Lieke Claassen, Thimo Jozef Theresia Boumans.

**Writing – review & editing:** Lieke Claassen, Patricia Maria Stassen, Dennis Gerard Barten, Marjolein Nel Tinie Kremers, Anne Maria Elisa Hermans, Noortje Zelis, Jochen Wilco Lennert Cals, Gideon Hubertus Petrus Latten.

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
